# Effects of *Lactobacillus plantarum* Postbiotics on Growth Performance, Immune Status, and Intestinal Microflora of Growing Minks

**DOI:** 10.3390/ani13182958

**Published:** 2023-09-19

**Authors:** Yalin Li, Shibo Zhen, Lin Cao, Fengxue Sun, Lihua Wang

**Affiliations:** College of Animal Science and Technology, Qingdao Agricultural University, Qingdao 266109, China; liyalin827@163.com (Y.L.); zhensbo@163.com (S.Z.); 13963906513@163.com (L.C.); sunfengxue2022@163.com (F.S.)

**Keywords:** postbiotics, lactobacillus plantarum, growth performance, immune status, intestinal microflora

## Abstract

**Simple Summary:**

The wet diet that minks have, due to its high moisture content, is prone to breeding harmful microorganisms in hot weather, resulting in severe diarrhea for minks. It has been proposed that probiotics can inhibit the intestinal colonization of pathogens that cause diarrhea. Probiotics as live microorganisms are beneficial in helping protect from chronic inflammatory bowel disease, improving intestinal flora structure and host immune function. However, it is not very convenient to maintain high viability of bacteria during processing, storage, and transportation of feeds with probiotics. Many studies have shown that inactivated probiotics and their metabolites also provide benefits for host health. These inactivated probiotics and their metabolites, known as postbiotics, have been accepted for their high safety and stability, easy storage and production, and long shelf life. This study proved that *Lactobacillus plantarum* postbiotics have biological effects on growing minks.

**Abstract:**

The present experiment was conducted to investigate the effects of *Lactobacillus plantarum* postbiotics on growth performance, immune status, and intestinal microflora of growing minks. A total of 80 minks (40 males and 40 females) were divided into four groups, each group contained 20 minks (10 males and 10 females). The minks in the four groups were fed a basal diet supplemented with 0, 0.15%, 0.3%, and 0.45% *Lactobacillus plantarum* postbiotics (PLP), respectively. After one week of adaptation, the experiment ran for eight weeks. The results showed that *Lactobacillus plantarum* postbiotics tended to have effects on average daily again (ADG) during the first 4 wk of the study (*p* < 0.1), and had effects on immune status (*p* < 0.05). *Lactobacillus plantarum* postbiotics also affected the abundance of intestinal bacteria at genus level (*p* < 0.05), but had no effects on α diversity of growing minks (*p* > 0.05). Compared to the minks in the control group, minks in 0.30% PLP group tended to have greater ADG, and IgA and IgM content in serum as well as SIgA content in jejunal mucosa (*p* < 0.05), and had less jejunal mucosal TNF-α and IL-8 levels, while minks in 0.45% PLP group had less IL-2 (*p* < 0.05). Compared to the control, *Lactobacillus plantarum* postbiotics decreased the relative abundances of *Bacteroides_vulgatus* and *Luteimonas_*sp. in male minks, and the relative abundances of *Streptococcus_halotolerans* in female minks (*p* < 0.05), respectively. Males grew faster and ate more associated with less F/G than females (*p* < 0.05). Males also had greater serum IgA and IgG content (*p* < 0.05), and males had less jejunal mucosal IL-1β, IL-8, IL-2, IL-6, IL-12, IL-10, TNF-α, and IFN-γ levels (*p* < 0.05). These results suggest that dietary supplementation of 0.3% postbiotics harvested from *Lactobacillus plantarum* could improve growth performance and immune status, and modulated the intestinal bacteria abundance of growing minks.

## 1. Introduction

Most minks are fed wet ranch diets containing more than 80% of fresh/frozen fish, egg, fish byproducts, and poultry byproducts. On hot days, these wet diets with high moisture content are susceptible to microbial spoilage resulting from contamination of *Salmonella*, Pathogenic *Escherichia coli*, etc. These spoiled diets may result in serious diarrhea in minks. It has been proposed that probiotics can inhibit the intestinal colonization of pathogens that cause diarrhea [1]. Probiotics as live microorganisms are beneficial in helping protect from chronic inflammatory bowel disease, improving intestinal flora structure and host immune function [2,3]. So, probiotics are microbiota-management tools for improving host health [2].

*Lactobacillus* have been used safely for years. Most *Lactobacillus* species are probiotic microorganisms and can improve gut health as a probiotics resource [4] because they produce organic acids (mainly lactic acid) [5] and enzymes with antibiotic properties. However, limited reports of *Lactobacillus*-associated infections have raised concerns about the safety of *Lactobacillus* [6]. In addition, many *Lactobacillus* are sensitive to oxygen and heat, so, providing strict storage and transportation conditions for these *Lactobacillus* is a challenge. Some studies have demonstrated that some postbiotics (known as metabolites, biogenic or cell-free supernatant soluble factors secreted by living bacteria or released by bacterial lysis) [7,8] exhibit bioactivities such as anti-inflammatory, immunomodulatory, antioxidant, and antimicrobial which are similar as probiotics [9]. Bacteriocins and SCFAs as postbiotics inhibit harmful bacterial species such as *E. coli* and *Salmonella* which are intolerant to acidic environments [10]. The soluble factors in postbiotics can promote cellular and humoral immunity and regulate the immune response, and affect the expression of anti-inflammatory factors and proinflammatory factors [11]. So, postbiotics can be used directly as active ingredients, without adhesion and colonization in the intestines, and can play a better and faster biological role [12]. Postbiotics are reasonably expected to have a more advantage than probiotics due to their clear chemical structure, long shelf life, higher safety and stability [13]. 

Postbiotics have been adopted rapidly in animal production in recent years. However, some aspects related to the potential health-promoting effects of postbiotics for minks remain unclear because of limited studies. The current experiment was conducted to evaluate the potential of postbiotics harvested from *Lactobacillus plantarum* to deliver benefits of immune responses and intestinal microflora of mink during their growing period.

## 2. Materials and Methods

The Animal Care and Use Committee of Animal Science and Technology, Qingdao Agricultural University reviewed and approved the experimental protocol and ethical clearance for this study.

### 2.1. Lactobacillus plantarum Probiotics

The *Lactobacillus plantarum* was purchased from North Carolina Bio. (No. 194165). *Lactobacillus plantarum* was activated so that the population of the viable count was more than 10^6–7^ cfu/mL according to plate counting. *Lactobacillus plantarum* in suspension was inactivated at 121 °C for 15 min. Inactivated *Lactobacillus plantarum* and milk powder were mixed at a ratio of 1:3 to prepare powder agents. The measurement of lactic acid in inactivated *Lactobacillus plantarum* suspension was 0.64%, and short-chain fatty acid was 0.06%. 

### 2.2. Animals, Facilities and Management

The present experiment was conducted in a commercial mink farm of Haiyang, Yantai. The study was started when minks were 12 wk old. A total of 80 minks of grey color type born in mid-April were selected from the farm’s animal stock. Minks that were selected for the study were weighed individually at the beginning of the experiment, at wk 4 and at wk 8 to obtain initial and final body weight, and average daily gain (ADG). The average daily feed intake (ADFI) was monitored for three days every week of the study on a cage basis to obtain the feed/gain ratio (F/G).

The minks were individually housed in identical wire mesh cages with dimensions of length (L) 75 cm, width (W) 30 cm, and height (H) 45 cm. Each cage was connected to a solid wooden nest box (30 cm × 30 cm × 30 cm, L × W × H) with a wire ceiling. Minks were allowed to enter and exit the nest box freely.

During the 8 wk period of the study, a wet diet was fed to meet the nutritional requirements of minks based on commercial standards. The wet diet manufactured by the farm was delivered on the top of the cages. The minks were fed approximately ad libitum with wet feed twice a day (6:30 a.m. and 5:30 p.m., respectively). Water was available in all cages ad libitum via a drinker. Animal health was checked twice daily throughout the study. When any minks were removed from the study due to poor health or compromised welfare, their body weights were recorded to adjust for ADG and F/G.

### 2.3. Experimental Design

A total of 80 minks (40 males and 40 females) were divided into 4 groups, each group contained 20 minks (10 males and 10 females). The minks in the four groups were fed a basal diet supplemented with 0, 0.15%, 0.3%, and 0.45% postbiotics of *Lactobacillus plantarum* (PLP), respectively. The experimental diets composition on as-fed basis and nutrient levels on air-dry basis are presented in Table 1.

### 2.4. Data Collection

#### 2.4.1. Lactic Acid and Short-Chain Fatty Acids

Samples of inactivated *Lactobacillus plantarum* suspension were thawed at 4 °C, 3 mL were taken from each EP tube, lyophilized, 1 mL of precooled methanol/acetonitrile/water (2:2:1, *v*/*v*/*v*) was added, mixed, ultrasounded in an ice bath for 1 h, stood at −20 °C for 2 h, and centrifuged 16,000× *g* at 4 °C for 20 min, and supernatant was obtained. The supernatant was dried in a high-speed vacuum enrichment centrifuge. During mass spectrometry, 400 μL methanol aqueous solution was added (1:1, *v*/*v*) redissolved, centrifuged 16,000× *g* at 4 °C for 20 min, and centrifuged through a C18 filter membrane 3000× *g* at 4 °C for 5 min. The supernatant was taken for analysis. During the whole analysis process, samples were placed in a 4 °C automatic injector and the SHIMADZU-LC30 ultra-high performance liquid chromatography system (UHPLC) was used. The ACQUITY UPLC^®^ HSS T3 (2.1 × 150 mm, 1.8 μm) (Waters, Milford, MA, USA) column was used. Positive (+) and negative (−) modes of each sample were detected by electrospray ionization (ESI). The samples were separated by UPLC for mass spectrometry using the QE Plus mass spectrometer (Thermo Scientific, Waltham, MA, USA) and ionization analysis using the HESI source.

#### 2.4.2. Immunity

Six males and six females (at 20 wk of age) in each group were selected to collect blood by heart punctures, and then euthanized to collect jejunal mucosal tissue at the end of the experiment. The blood was centrifuged at 4 °C for 10 min at 3000 r/min, and the serum was obtained. The contents of IgA, IgM, and IgG in serum were determined. Jejunal mucosal tissue was taken, and 9 mL normal saline was added to make tissue homogenate. The tissue was centrifuged at 4 °C for 10 min at 3500 r/min, and the supernatant was obtained. The concentrations of secreted immunoglobulin A (SIgA), interleukin-1β (IL-1β), interleukin-8 (IL-8), interleukin-10 (IL-10), interleukin-2 (IL-2), interleukin-6 (IL-6), interleukin-12 (IL-12), tumor necrosis factor-α (TNF-α), and interferon-γ (IFN-γ) were determined. All these indicators were measured using enzyme-linked immunosorbent assay (ELISA) kits (Nanjing Jiancheng Biological Engineering Research Institute, Nanjing, China), and all of them were determined by a full-wavelength micro-plate reader (Tecan Austria GmbH, Groedig, Austria).

#### 2.4.3. Intestinal Flora

Rectal content samples were collected at the same time as the jejunal mucosal tissue collection at the end of the experiment. DNA from rectal contents was extracted using the Fast DNA Spin Kit for Soil kit (MP, Santa Ana, CA, USA). The extracted genomic DNA was detected by 1% agarose gel electrophoresis. The primers 338F (5′-ACTCCTACGGGAGGCAGCAG-3′) and 806R (5′-GGACTACHVGGGTWTCTAAT-3′) on an ABI GeneAmp PCR system 9700 thermal cycler were used to amplify the V3-V4 region of the DNA [14]. The PCR amplification reaction program included the following: pre-denaturation at 95 °C for 3 min, for a total 27 cycles (denaturation at 95 °C for 30 s, annealing at 55 °C for 30 s, and extension at 72 °C for 30 s), and then final extension at 72 °C for 10 min. Then, the amplicons were excised from the 2 % agarose gel, purified using the AxyPrep DNA Gel Extraction Kit (Axygen, Union City, CA, USA), and tested by 2% agarose gel electrophoresis. Quantification was performed using the QuantiFluor™-ST Blue fluorescence quantification system (Promega, Madison, WI, USA). A PE 300 library was constructed based on the Illumina MiSeq platform and sequenced using the Illumina MiSeq PE 300 platform [15].

### 2.5. Statistical Analysis

The data of growth performance and immunity indictors were analyzed using the general linear model of SPSS 25.0 software (SPSS Inc., Chicago, IL, USA, GLM). The model included *Lactobacillus plantarum* postbiotics, gender, and their interactions as fixed effects. The differences between means were tested by Duncan’s multiple comparisons. Significant differences were identified at *p* < 0.05 and trends at *p* < 0.10.

The original sequencing sequence of bacteria was spliced using Trimmomatic 1.2.11 software quality control and FLASH 1.2.11 software. The optimized sequences were clustered by operational classification unit (OTU) according to 97% similarity, and the flora composition of each sample at various levels was statistically analyzed [15]. Spearman correlation coefficient was used for correlation analysis. The test results were analyzed on the I-Sanger cloud platform.

## 3. Results

### 3.1. Growth Performance

Lactobacillus plantarum postbiotics tended to have effects on ADG during the first 4 wk of the study (*p* < 0.1; Table 2), and had no effects on the other performance (*p* > 0.05). Compared to the minks in control, minks in the 0.3% PLP group tended to have greater ADG (*p* < 0.1).

Males were heavier than females at the start of the study at wk 0 (*p* < 0.001), and at the end of the study at wk 8 (*p* < 0.001). Compared to females, males had greater ADG, ADFI, and less F/G during the first 4 wk, the last 4 wk, and the entire 8 wk period of the study (all *p* < 0.05). Interactions between gender and *Lactobacillus plantarum* postbiotics for ADG during the first 4 wk of the study tended to be significant (*p* < 0.1), males from the 0.3% PLP group had greater ADG than males in the other groups (*p* < 0.05), but ADG of females from the 0.3% PLP group was not different from females in the other groups (*p* > 0.05).

### 3.2. Immune Function

#### 3.2.1. Serum Immune Indexes

*Lactobacillus plantarum* postbiotics had significant effects on serum IgA and IgM concentrations (*p* < 0.05; Table 3), and had no significant effects on IgG concentration (*p* > 0.05). Compared to the control, the 0.30% PLP group had significantly increased IgA and IgM concentrations in serum (*p* < 0.05). Males had greater serum IgA and IgG concentrations than females (*p* < 0.05). Interactions between Lactobacillus plantarum postbiotics and gender were not evident for serum immune indexes measured (*p* > 0.05).

#### 3.2.2. Mucosal Immune Indexes

*Lactobacillus plantarum* postbiotics had effects on SIgA, IL-8, IL-2, TNF-α, and IFN-γ levels in jejunal mucosa (*p* < 0.05; Table 4), and had no effects on IL-1β, IL-6, IL-12, and IL-10 levels (*p* > 0.05). Compared to the control, the 0.30% PLP group had increased SIgA and decreased TNF-α, IL-8 levels, while the 0.45% group had decreased IL-2 (*p* < 0.05). Males had lower jejunal mucosal IL-1β, IL-8, IL-2, IL-6, IL-12, IL-10, TNF-α, and IFN-γ levels than females (*p* < 0.05). Interactions between *Lactobacillus plantarum* postbiotics and gender were significant for jejunal mucosa IL-10 and TNF-α levels (*p* < 0.05).

### 3.3. Intestinal Flora

Interactions of *Lactobacillus plantarum* postbiotics and gender were not evident for rectal microbiota abundance and composition (all *p* > 0.05). So, the main effects of *Lactobacillus plantarum* postbiotics and gender on rectal microbiota richness are presented (Figure 1, Figure 2 and Figure 3).

#### 3.3.1. Intestine Microbiome of Male Minks

The abundance of rectal flora estimated by Shannon, Simpson, Chao1, and Ace indices for male minks was not significant (all *p* > 0.05; Figure 1a). The data obtained confirmed that *Lactobacillus*, *Lactococcus*, *Enterococcus*, and an unknown member of *Peptostreptococcaceae* were the most abundant genus for the control group (Figure 1b). *Escherichia-Shigella*, an unknown member of *Butyricicoccaceae*, *Lactobacillus,* and *Enterococcus* were the most abundant genus for the 0.15% PLP group. *Lactobacillus*, *Streptococcus*, *Cetobacterium* and *Escherichia-Shigella* were the most abundant genus for the 0.30% PLP group. *Streptococcus*, *Candidatus_Arthromitus*, *Enterococcus*, and *Cetobacterium* were the most abundant genus for the 0.45% PLP group. Compared to the control, the PLP groups decreased the relative abundances of *Bacteroides_vulgatus* and *Luteimonas_*sp. (*p* < 0.05; Figure 1c).

#### 3.3.2. Intestine Microbiome of Female Minks

The abundance of rectal flora estimated by Shannon, Simpson, Chao1, and Ace indices for female minks was not significant (all *p* > 0.05; Figure 2a). The data obtained confirmed that *Streptococcus*, *Lactobacillus*, *Weissella,* and *Staphylococcus* were the most abundant genus for the control group (Figure 2b). *Escherichia-Shigella*, *Streptococcus*, *Fusobacterium,* and *Enterococcus* were the most abundant genus for the 0.15% PLP group. *Lactobacillus*, *Streptococcus*, *Staphylococcus* and an unknown member of *Peptostreptococcaceae* were the most abundant genus for the 0.30% PLP group. *Streptococcus*, an unknown member of *Peptostreptococcaceae*, *Weissella,* and *Lactobacillus* were the most abundant genus for the 0.45% PLP group. Compared to the control, the PLP groups decreased the relative abundance of *Streptococcus_halotolerans* (*p* < 0.05; Figure 2c).

#### 3.3.3. Correlation Analysis

The results for the spearman relation heatmap are shown in Figure 3. *Cetobacterium* and *Candidatus_Arthromitus* were positively correlated with IL-8, TNF-α, IL-1β, and IL-12 (*p* < 0.05). *Cetobacterium* and *Candidatus_Arthromitus* had a negative correlation with IgG (*p* < 0.05). *Candidatus_Arthromitus* had a negative correlation with IgA (*p* < 0.05). An unknown member of *Peptostreptococcaceae* had a negative correlation with IL-8, IL-10, IL-12, IL-1β, and SIgA (*p* < 0.05). *Weissella* had positive correlation with IgG (*p* < 0.05), and negative correlation with TNF-α and IL-12 (*p* < 0.05). *Fusobacterium* had a negative correlation with IFN-γ and TNF-α (*p* < 0.05). *Norank_f_norank_o_hloroplast* and an unknown member of *Butyricicoccaceae* had a negative correlation with TNF-α (*p* < 0.05).

## 4. Discussion

Results of the current study indicate that *Lactobacillus plantarum* postbiotics could improve growth performance in the early stage of growing minks. This agrees with the results of previous studies that *Lactobacillus plantarum* postbiotics increased ADG and decreased F/G in the early stage of broilers [16,17], and improved the growth performance of weaned piglets [18]. This study proved that the postbiotics harvested from *Lactobacillus plantarum* possesses growth promoting effect on growing minks. The potential of *Lactobacillus plantarum* postbiotics is ascribed to the secondary metabolites of the *Lactobacillus plantarum*, including the organic acids, teichoic acid (TA), peptidoglycan (PGN), bacteriocin proteins, exopolysaccharides (EPS), neurotransmitters, and antioxidant compounds [19]. Organic acids could reduce the intestinal pH value associated with inhibition of pathogens, and promotion of intestinal development, which could contribute to intestinal mucosal barrier function, improve intestinal microecological balance, and promote utilization of nutrients, and consequently, improve the growth performance of animals [19,20].

However, it appears that the improved ADG of minks from the 0.3% *Lactobacillus plantarum* postbiotics group were only evident for the initial 4 wk of the study. The result suggests short-term effects of *Lactobacillus plantarum* postbiotics on growth performance. In addition, the digestive tract of minks develops slowly [21]. Compared to females, male minks eat more and grow faster associated with a sensitive digestive tract. This could explain why *Lactobacillus plantarum* postbiotics have beneficial effects for males in the study. Unlike other domestic animals, it is more important for minks to ensure optimum body size (not only body weight) and health status for quality at pelting.

Immunoglobulins (IgA, IgG, and IgM) have antibacterial and antiviral effects as an important part of the humoral immune system of animals. IgM regulate subsequent immune responses, facilitating the production of IgG and the initial response against exotic antigens [22]. IgA affects a variety of protective mechanisms in the body through its interaction with different receptors and immune mediators [23]. In this experiment, *Lactobacillus plantarum* postbiotics increased the IgA content in serum. IgA protects mucosal surfaces by preventing the entry and colonization of toxins and pathogens [23].

SIgA regulates mucosal homeostasis and protects the intestinal epithelial cells from bacterial toxins [24]. SIgA protects the mucosal surface in the mucosa by presenting the bacterial antigens to dendritic cells [25]. The results of this experiment showed that the content of SIgA increased with the increase of postbiotics supplementation. Products shed by members of the microbial community promote production of SIgA and polymeric immunoglobulin receptors by activating pattern recognition receptors on host epithelial and immune cells [26]. The intestinal mucosal immune barrier is the coordination between various proinflammatory cytokines and anti-inflammatory cytokines to regulate the occurrence and development of inflammation [27]. Compounds of the cell wall might boost the immunological system [28,29,30]. Chiu’s study showed that dietary *Lactobacillus plantarum* could inhibit the expression of IL-1β and TNF-α in the jejunal mucosa as well as IL-8 in the jejunum and ileum mucosa [31]. In this study, the expression of TNF-α was decreased, which might be related to TLR2. TLR2 can recognize peptidoglycan and lipoteichoic acid of Gram-positive bacteria. Metabolites of *Lactobacillus plantarum* are recognized by TLR2. Reduction of TLR2 increases susceptibility to intestinal injury and inflammation, and TLR2 stimulation can effectively improve intestinal epithelial barrier function [32]. Lipoteichoic acid (LTA) regulates NF-κB-mediated proinflammatory responses in the intestinal tissue environment [33,34]. By inhibiting NF-κB signaling, the expression of proinflammatory cytokines such as IL-6, IL-1β, and TNF-α is decreased [35]. The results showed that the content of INF-γ decreased, which might be related to TLR9. These PAMPs act as activators of the endosomal TLR9, present in multiple immune cells (e.g., DCs and B cells), IECs, and various lung cells [36,37,38]. TLR9 is especially engaged in shifting immunological responses between Th1 (INF-γ producing) and Th2 (mainly producing IL-4) lymphocytes and in the maintenance of immunological homeostasis in the intestines [39]. These results indicate that *Lactobacillus plantarum* postbiotics can regulate immunity and reduce intestinal inflammatory responses. In addition, our results indicate that SIgA increased quadratically, while IL-8, TNF-α, and IFN-γ decreased quadratically with the increasing supplementation of *Lactobacillus plantarum* postbiotics. Results of the current study suggest that proper supplementation of *Lactobacillus plantarum* postbiotics could improve immunity. The proper supplementation of *Lactobacillus plantarum* postbiotics was determined to be 0.23% and 0.37% according to quadratic regression of the IL-8 and SIgA, respectively.

A stable and balanced microbiome in the gut is necessary for optimal health and productivity. The gut microbiota is thought to regulate the host immune system [40]. This study confirmed that *Lactobacillus plantarum* postbiotics had no effects on Alpha diversity at a genus level for males and females. In the current study, the most abundant genus for males were *Streptococcus*, *Lactobacillus*, *Escherichia*-*Shigella,* and an unknown member of *Peptostreptococcaceae*, and the most abundant genus for females were *Lactobacillus, Cetobacterium, Streptococcus,* and *Enterococcus*, respectively. Previous studies suggested that *Bacteroides_vulgatus* may be implicated in the pathogenesis of ulcerative colitis [41,42]. *Luteimonas_*sp. is a newly discovered strain derived from coral mucus. The primary sequence of *Luteimonas_*sp. contained the BpeGReg and EcGReg/DosC domains from *Bordetella pertussis* and *Escherichia coli*, respectively [43]. These two bacteria have not been well studied in animals. The reduction of *Luteimonas_*sp. may be related to the EPS produced by *Lactobacillus plantarum*. EPS can prevent harmful bacteria from adhering to intestinal epithelial cells and promote the proliferation of beneficial bacteria, and then reduce harmful bacteria and harmful substances, and can, consequently, improve the diversity and balance of intestinal microorganisms [44]. In agreement with previous studies, postbiotics harvested from *Lactobacillus plantarum* increased the *Lactobacillus* population and inhibited *Escherichia coli* [45,46,47]. The relative abundance of *Lactobacillus* and *Cetobacterium* in the 0.30% group was significantly increased, while the relative abundance of *Enterococcus* was decreased. Aggregation-promoting factor, a metabolite of *Lactobacillus plantarum,* has a probiotic effect on *lactobacillus* colonization and the establishment of beneficial bacteria dominant flora in the intestinal tract [48,49]. *Cetobacterium* is an anaerobic bacterium present in the gut of fish [50]. The diet of mink contained 30% fish products, so, *Cetobacterium* was detected in the gut of mink. *Cetobacterium* was a key anaerobic indigenous gut bacterium for maintaining host health by regulating the synthetization of B12 [51]. *Enterococcus* is a pathogenic bacterium that spread horizontally between humans and animals and has a certain drug resistance [52,53]. *Streptococcus*_*halotolerans* were significantly reduced in female minks. *Streptococcus*_*halotolerans* is a member of *Streptococcus*, and shares 40.5% homology with *Streptococcus Thoraltensis* [54]. The effects and mechanisms of *Enterococcus* and *Streptococcus_halotolerans* are not still clear, so, a more intensive study is needed on them. The intestinal microbiota interacts with immunity [55]. Xu reported that *Lactobacillus* spp. were positively correlated with IgG and IgA [56]. However, our study did not obtain the correlation between *Lactobacillus* spp. and IgA and IgG, perhaps due to the animals being different.

## 5. Conclusions

The results obtained in this study showed that dietary supplementation of 0.3% postbiotics harvested from *Lactobacillus plantarum* could improve growth performance and immune status and modulated the intestinal bacteria abundance of growing minks. The proper supplementation of *Lactobacillus plantarum* postbiotics was determined to be 0.23% and 0.37% according to quadratic regression of the IL-8 and SIgA, respectively.

## Figures and Tables

**Figure 1 animals-13-02958-f001:**
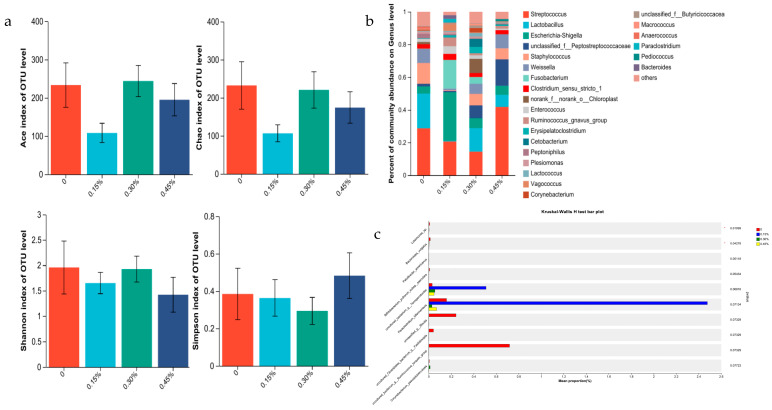
(**a**) Alpha diversity indices of mal e minks at growing stage (Shannon index, Simpson index, Ace index, and Chao index). (**b**) Mean relative abundance of genus by male minks. (**c**) A species of bacteria showing differences in abundance of microbial communities in male minks.

**Figure 2 animals-13-02958-f002:**
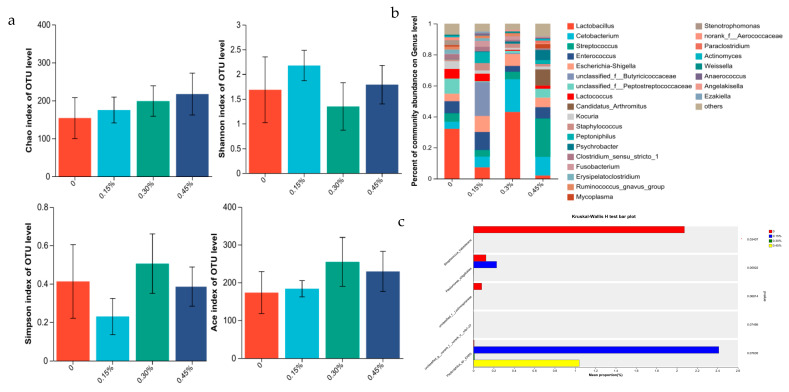
(**a**) Alpha diversity indices of female minks at growing stage (Shannon index, Simpson index, Ace index, and Chao index). (**b**) Mean relative abundance of genus by female minks. (**c**) A species of bacteria showing differences in abundance of microbial communities in female minks.

**Figure 3 animals-13-02958-f003:**
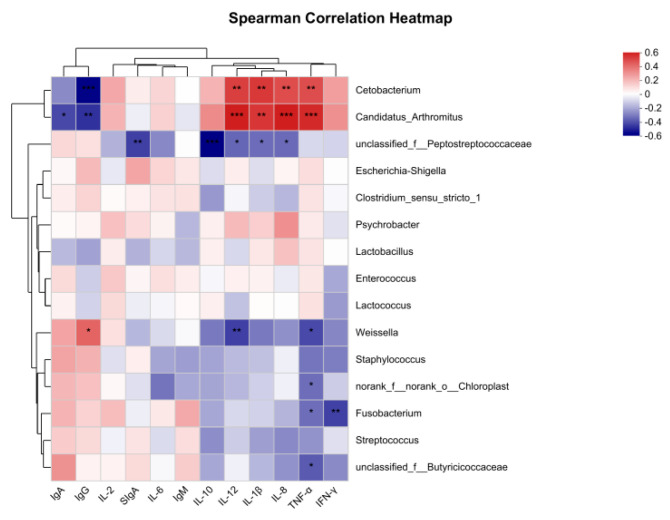
Heatmap of intestinal flora and immune correlation in minks (at genus level). The X-axis and Y-axis are environmental factors and species, respectively, and the correlation R- and *p*-values are obtained through calculation. R-values are shown in different colors in the figure. If *p*-values are less than 0.05, they are marked by *. The legend on the right is the color range of different R-values. * 0.01 < *p* ≤ 0.05, ** 0.001 < *p* ≤ 0.01, *** *p* ≤ 0.001.

**Table 1 animals-13-02958-t001:** Ingredient composition (as-fed basis) and chemical composition (air-dry basis) of basal diets %.

Items	Content
Sea fishes	30.00
Unhatched fertilized egg	24.00
Chicken ribs	10.00
Chicken head	15.00
Extruded corn	7.00
Chicken livers	5.00
Wheat bran	3.00
Spary-dried blood cells	2.00
Soybean meal	3.00
Premix ^1^	1.00
Total	100.00
Nutrient levels	
DM	41.91
ME (MJ/kg) ^2^	16.59
Ether extract	21.55
Crude protein	34.66
Calcium	1.10
Phosphorus	1.21

^1^ The premix provided the following per kg of the diet: VA 9000 IU, VC 40 mg, VE 20 mg, VK_3_ 0.5 mg, VB_1_ 5 mg, VB_2_ 3 mg, VB_6_ 2.5 mg, VB_12_ 1 mg, VD_3_ 2000 IU, nicotinic acid 20 mg, pantothenic acid 6 mg, folic acid 0.5 mg, biotin 0.5 mg, Fe 30 mg, Zn 25 mg, Mn 10 mg, Cu 5 mg, I 0.25 mg, and Se 0.2 mg. ^2^ The value of metabolizable energy is the calculated value, and the other value is the actual measured value. ME is calculated using the equation ME = (0.85 × CP% × 4.5 + 0.90 × EE% × 9.5 + 0.75 × NFE% × 4.0) × 4.184, NFE(%) = 100% − CP(%) − EE(%) − Ash(%).

**Table 2 animals-13-02958-t002:** Effects of postbiotics on growth performance of growing mink.

	PLP				SEM	Gender		SEM	*p*-Value		
	0	0.15%	0.30%	0.45%		Male	Female		*P* _PLP_	*P* _Gender_	*P* _Gender×PLP_
Wt/g											
Initial (wk 0)	1099.97	1128.10	1117.87	1122.38	32.65	1310.67	923.48	23.85	0.94	<0.001	0.64
Final (wk 8)	1859.02	1967.36	2026.89	1941.19	68.22	2394.49	1502.74	48.43	0.36	<0.001	0.58
ADG, g											
Wk 0 to wk 4	15.41	15.67	19.32	15.71	1.18	20.39	12.66	0.84	0.06	<0.001	0.05
wk 4 to wk 8	11.70	14.30	13.15	13.54	1.22	18.31	8.03	0.83	0.47	<0.001	0.95
Wk 0 to wk 8	13.55	14.99	16.23	14.62	0.89	19.35	10.34	0.62	0.17	<0.001	032
ADFI, g											
wk 0 to wk 4	283.76	306.49	302.57	294.90	10.74	335.37	258.49	7.37	0.42	<0.001	0.71
wk 4 to wk 8	284.99	296.86	297.36	292.27	9.54	343.36	242.38	6.54	0.75	<0.001	0.99
wk 0 to wk 4	284.37	301.67	299.96	293.58	9.31	339.36	250.43	6.38	0.51	<0.001	0.91
G:F											
wk 0 to wk 4	20.32	20.76	17.61	20.27	1.35	17.57	21.91	0.93	0.30	0.002	0.38
wk 4 to wk 8	29.58	24.75	27.00	25.69	2.14	20.71	32.80	1.47	0.38	<0.001	0.94
wk 0 to wk 8	22.63	21.89	20.36	21.56	0.98	18.10	25.12	0.69	0.41	<0.001	0.68

**Table 3 animals-13-02958-t003:** Effects of different levels of postbiotics on serum immune indexes of mink.

	PLP				SEM	Gender		SEM	*p*-Value		
	0	0.15%	0.30%	0.45%		Male	Female		*P* _PLP_	*P* _Gender_	*P* _Gender×PLP_
IgA/(μg/mL)	328.10 ^b^	346.33 ^ab^	360.42 ^a^	343.85 ^ab^	7.72	361.52	327.83	4.78	0.005	<0.001	0.18
IgM/(μg/mL)	1569.73 ^b^	1712.11 ^ab^	1914.27 ^a^	1496.05 ^b^	81.29	1719.12	1626.96	65.05	0.003	0.24	0.07
IgG/(g/L)	46.38	45.07	42.68	43.76	2.52	51.29	37.65	1.08	0.37	<0.001	0.57

Means within a row and main effect with unlike superscripts differ (*p* < 0.05).

**Table 4 animals-13-02958-t004:** Effects of different levels of postbiotics on mucosal immune indexes of growing mink.

	PLP				SEM	Gender		SEM	*p*-Value		
	0	0.15%	0.30%	0.45%		Male	Female		*P* _PLP_	*P* _Gender_	*P* _Gender×PLP_
SIgA/(ng/mL)	2105.11 ^b^	2234.11 ^b^	2370.64 ^a^	2229.69 ^b^	45.75	2241.54	2228.23	37.22	0.004	0.78	0.91
IL-1β/(pg/mL)	228.35	229.31	226.17	218.09	9.68	200.85	250.10	4.06	0.62	<0.001	0.85
IL-8/(pg/mL)	129.29 ^a^	124.81 ^a^	111.10 ^b^	125.97 ^a^	6.14	107.20	138.39	3.17	0.02	<0.001	0.59
IL-2/(pg/mL)	314.12 ^a^	304.31 ^a^	289.14 ^ab^	251.76 ^b^	17.23	255.21	324.45	10.88	0.02	<0.001	0.82
IL-6/(pg/mL)	31.96	28.27	29.21	29.26	1.82	27.01	32.34	1.11	0.47	0.003	0.95
IL-12/(pg/mL)	22.30	23.13	21.97	21.28	1.34	18.82	25.53	0.61	0.54	<0.001	0.47
IL-10/(pg/mL)	92.18	88.94	91.57	86.62	3.94	81.54	98.11	2.33	0.50	<0.001	0.002
TNF-α/(pg/mL)	992.95 ^a^	799.21 ^b^	784.09 ^b^	892.03 ^ab^	99.96	576.28	1157.86	36.58	0.01	<0.001	0.01
IFN-γ/(pg/mL)	738.87 ^ab^	698.74 ^ab^	635.42 ^b^	819.48 ^a^	46.43	619.48	826.78	28.53	0.01	<0.001	0.07

Means within a row and main effect with unlike superscripts differ (*p* < 0.05).

## Data Availability

Not applicable.

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
