# Peer review of "Effects of Lactobacillus plantarum Postbiotics on Growth Performance, Immune Status, and Intestinal Microflora of Growing Minks"

_animals, 2023, doi:10.3390/ani13182958_

Round 1

Reviewer 1 Report

Comments to the Authors of manuscrip ID animals-2563090 , entitled ,,Effects of lactobacillus plantarum postbiotics on growth performance, immune status and intestinal Microflora of growing minks’’.

The manuscript discusses the influence of postbiotic derived from Lactobacillus Plantarum Probiotics on growth performance, levels of immunoglobulins  and intestinal micloflora of growing minks. The experiment was carried out on a group of 40 breeding  american minks (neogale vison) in farm conditions. The duration of the study was 8 weeks. Each of the three research groups consisted of 10 females and 10 males. Postbiotics in three different doses were administered to individual research groups with food every day. The control group consisted of minks that did not receive the postbiotics. Various measures, including growth performance, IgA, IgM, IgG content in serum and SIgA,TNF-α, IL-8 ,IL-2, IL-1β, IL-8, IL-2, IL-6, IL-12, IL-10, TNF-α, IFN-γ levels content in jejunal mucosa were conducted The obtained results showed that postbiotics supplementation has a beneficial effect on innate immune system, humoral immunity and mink growth.

The quality of the obtained postbiotics:  

The composition of the obtained postbiotic is not given. What was the active ingredient of the postbiotic? Whether the composition of the postbiotic has been examined? The fact that Lactobacillus plantarum are resistant to high temperatures is well known but deactivation process seems to be too intense.  The temperature of 120 C for 15 minutes could cause denaturation of immunogenous bacterial compounds and lead to the destruction of products of probiotic bacteria such as butyric acid. Where did the method of obtaining postbiotics from bacteria come from? Please provide literature. Please discuss.

Blood sample collection:

Blood collection  through cardiac puncture is a more stressful method than taking blood through an incision of the claw. Whether the amount of blood collected from claw is insufficient to determine the levels of IgG and IgM in the ELISA method? Please discus

  1. L 80: italics unnecessary in the word prebiotics.
  2. L 25 no explanation of the ADG shortcut. The explanation of the shortcut should be in the abstract and not only in the materials and methods.
  3. L 22: ,, replicants’’ -the word does not fit the context of the sentence that contains information about the number of minks, ,, replicants’’ may be used elsewhere to describe mink’s intenstines homogenate.
  4. Identification of bacterial species of the intestinal microbiota was carried out in the last stage of the experiment, which is a big loss for the value of the work. Taking swabs from the rectum or isolating bacteria from feces multiple times during the 8-week experiment would make it possible to track changes in the mink intestinal microbiota, also providing information on the effect of the postbiotic on the mink intestinal microbiota.
  5. Noteworthy is the very good statistical presentation of the obtained results, however, more data should be discussed in the discussion, moreover he discussion contains too much commonly known information about the beneficial effects of antibiotics on the animal's body.
  6. The serum level of immunoglobulins should also be determined before administration of postbiotics, which would make it possible to know the initial level of IgA, IgG, IgE.

Author Response

September 11th, 2023

Subject: Revision of Manuscript (ID: animals -2563039) entitled: “Effects of Lactobacillus plantarum Postbiotics on Growth 2 Performance, Immune Status and Intestinal Microflora of 3 Growing Minks”

Dear reviewer,

Thank you for spending your value time on our manuscript. We appreciated your fine comments. We have revised the manuscript based on your comments. Attached is the revision of the manuscript.

Our specific responses to your comments (in italics) follow:

The composition of the obtained postbiotic is not given. What was the active ingredient of the postbiotic? Whether the composition of the postbiotic has been examined? The fact that Lactobacillus plantarum are resistant to high temperatures is well known but deactivation process seems to be too intense.  The temperature of 120 C for 15 minutes could cause denaturation of immunogenous bacterial compounds and lead to the destruction of products of probiotic bacteria such as butyric acid. Where did the method of obtaining postbiotics from bacteria come from? Please provide literature. Please discuss.

Response: I added the product information in lines 84-86, and the analysis method has been provided in 2.4.1. Lactic acid and short-chain fatty acids.

Blood collection  through cardiac puncture is a more stressful method than taking blood through an incision of the claw. Whether the amount of blood collected from claw is insufficient to determine the levels of IgG and IgM in the ELISA method? Please discus

 Response: In previous experiment, we took only 0.6-0.8 ml blood through an incision of the claw, it may be sufficient to determine IgG, IgA and IgM. However, we also measured GSH-PX, MDA, T-AOC and other serum indicators (these results were reported separately). We required more than 3 ml blood sample, so we have to take blood through cardiac puncture.

L 80: italics unnecessary in the word prebiotics.

Response: According to the template, italics is necessary for the word prebiotics in the sub-title.

L 25 no explanation of the ADG shortcut. The explanation of the shortcut should be in the abstract and not only in the materials and methods.

Response: Explanation of the ADG shortcut was added as recommended (Line 25 and Line 92).

L 22: ,, replicants’’ -the word does not fit the context of the sentence that contains information about the number of minks, ,, replicants’’ may be used elsewhere to describe mink’s intenstines homogenate.

Response: Corrections have been made in Line 22 and Line 108.

Identification of bacterial species of the intestinal microbiota was carried out in the last stage of the experiment, which is a big loss for the value of the work. Taking swabs from the rectum or isolating bacteria from feces multiple times during the 8-week experiment would make it possible to track changes in the mink intestinal microbiota, also providing information on the effect of the postbiotic on the mink intestinal microbiota.

Response: Thank you very much for your advice. It is a good idea to track changes in the mink's gut microbiota through taking swabs from the rectum or isolating bacteria from feces. We will take the advice in future studies, but to track changes in the mink intestinal microbiota is out of the scope of the current study.

Noteworthy is the very good statistical presentation of the obtained results, however, more data should be discussed in the discussion, moreover he discussion contains too much commonly known information about the beneficial effects of antibiotics on the animal's body.

Response: In the reversion, we make the discussion more objective ( 4. Discussion ).

The serum level of immunoglobulins should also be determined before administration of postbiotics, which would make it possible to know the initial level of IgA, IgG, IgE.

Response: It is much better to know the initial level of IgA, IgG, IgE. However, there was a control (only fed the basic diet) in the experimental, the current results could illustrate the effects postbiotics.

I hope our revision has addressed all the comments from you. Please let me know if you have any questions or further comments. Again, thank you for your value time!

Regards,

Yalin Li

College of Animal Science and Technology

Qingdao Agricultural University

Shandong, China

Reviewer 2 Report

This is a good study, but limiting in some detail. Please see comments in text.

This is good study, but does not provide substantial background on the research problem. Some comments are given on the document.

Author Response

September 11th, 2023

Subject: Revision of Manuscript (ID: animals -2563039) entitled: “Effects of Lactobacillus plantarum Postbiotics on Growth 2 Performance, Immune Status and Intestinal Microflora of 3 Growing Minks”

Dear reviewer,

Thank you for spending your value time on our manuscript. We appreciated your fine comments. We have revised the manuscript based on your comments. Attached is the revision of the manuscript.

Our specific responses to your comments (in italics) follow:

1 week of adaptation and the experiment ran for 8 weeks. I assume? Please be clear.

Response: The minks in the present experiment were individually housed, while these minks were housed in group of 3 prior to the experiment. So, after one week of adaptation, the experiment began. Corrections have been made (Line 24).

Any specific pathogens?

Response: Some specific pathogens have been added as recommended (Line 44-47).

Including ethical clearance?

Response: Yes, ethical clearance was added to clarify (Line 77-78).

Was weight recorded at the start and as the study progressed? How often were the measurements taken and how?

Response: Yes, Line 88-94 were added to address this issue.

For what purpose?

Response: Minks were allowed to enter and exit the nest box freely. The enriched cages can improve mink welfear.

Were allocated feed quantities and intake recorded? How and at what intervals?

Response: Yes, we recorded feed intake and monitored the average daily feed intake (ADFI) for three days per week during the study period. Line 88-94 were added to address this issue.

Was chemical testing done on individual ingredients before mixing? Especially, the animal protein, any quality testing done?

Response: In fact, chemical testing and quality testing were not done on individual ingredients. We just analyzed the chemical composition of diets.

Perhaps this should also be stated under 2.2 - animal management.

Response: Corrections have been made as recommended (2.2 - animal management).

Presumably, week 0 refers to the start of the experiment, at placement, since the animals were 10 weeks old?

Response: Corrections have been made (Line 90-94), .We tried to describe clearly.

If the experiment started when the minks were 10 weeks old, at what age were these measurements taken? Please be specific.

Response: According the corrections made, these minks were 20-wk old at the end of experiment. The information of age was added (Line 138-140).

Was this a negative (no additive to protect against pathogens in feed) or positive control (a standard additive to protect against pathogens in feed was used)??

Response: No additive to protect against pathogens in feed was used in control.

Was this across treatments?

Response: No. We have deleted line 246 and made corrections accordingly.

The Discussion is too general and focuses less on the reported results in this study. Any speculation on the microbial colonization noted with different treatments.

Response: We have revised the discussion of the flora part ( 4. Discussion Line333-366 ).

Please elaborate on how you arrived at this conclusion. The discussion is limiting in this detail.

Response: We tried to elaborated in detail in the discussion, and we added quadratic regression in discussion (Line326-332).

I hope our revision has addressed all the comments from you. Please let me know if you have any questions or further comments. Again, thank you for your value time!

Regards,

Yalin Li

College of Animal Science and Technology

Qingdao Agricultural University

Shandong, China

Reviewer 3 Report

Brief Summary

This article is generally well done and written in correct English.

The study is well structured, the appropriate design, the statistics and the techniques used, seem as far as I can judge to be well suited. The areas covered in zootechnics, immunity and microbiology are complete.

Points need to be reviewed, particularly in the area of analysis of results

Concerning the general approach, my only reservation also concerns the publishers. Despite the scientific qualities of the article, the production of mink for pelting is now prohibited in certain countries in Europe. Is this article in agreement with the editorial line of the newspaper?

General concept

It would have been good to provide more information regarding the composition of the product tested; simple parameters could have been measured such as lactic acid. Please provide details in material and methods.

I have concerns about the presentation of the results, considering that the Treatment x Gender interaction is significant or at the limit of significance for the growth of the first week, of certain parameters of mucosal immunity, I would advise to separate the presentation of results between males and females as for the microbiota.

In the discussion a point is missing concerning the quadratic aspect of the results. In fact, the results at the maximum dose of postbiotic are lower than those at the 0.3% dose. Please provide technical elements regarding this difference.

Specific comments referring to line numbers,

Line 23 Lactobacillus instead Lctbacillus

Lines 56-58 In this passage I understand that there are some very limited risks of using live bacterial probiotics but I find the wording unclear and not explicit can you clarify please.

Lines 106-107 please put “Nutrient levels” in bold.

Line 271 “for quality” instead of “fur quality”

Line 310 “which can effectively reduce the pH value in the intestine” have you measured the intestinal pH or that of the faeces ? again it would have been interesting to measure the quantity of lactic acid in the product.

Author Response

September 11th, 2023

Subject: Revision of Manuscript (ID:animals -2563039) entitled: “Effects of Lactobacillus plantarum Postbiotics on Growth 2 Performance, Immune Status and Intestinal Microflora of 3 Growing Minks”

Dear reviewer,

Thank you for spending your value time on our manuscript. We appreciated your fine comments.  We have revised the manuscript based on your comments. Attached is the revision of the manuscript.

Our specific responses to your comments (in italics) follow:

It would have been good to provide more information regarding the composition of the product tested; simple parameters could have been measured such as lactic acid. Please provide details in material and methods.

Response: I added the product information in lines 84-86, and the analysis method has been  provided in 2.4.1. Lactic acid and short-chain fatty acids

I have concerns about the presentation of the results, considering that the Treatment x Gender interaction is significant or at the limit of significance for the growth of the first week, of certain parameters of mucosal immunity, I would advise to separate the presentation of results between males and females as for the microbiota.

Response: We totally agree with the reviewer’s comments. The results of microbiata have presented separately for males and females.

In the discussion a point is missing concerning the quadratic aspect of the results. In fact, the results at the maximum dose of postbiotic are lower than those at the 0.3% dose. Please provide technical elements regarding this difference.

Response: we added quadratic regression in discussion (Line326-332).

Line 23 Lactobacillus instead Lctbacillus.

Response: Corrections have been made as recommended (Line 23).

Lines 56-58 In this passage I understand that there are some very limited risks of using live bacterial probiotics but I find the wording unclear and not explicit can you clarify please.

Response: Corrections have been made in line55-56 to clarify

Lines 106-107 please put “Nutrient levels” in bold.

Response: Corrections have been made as recommended (Line 111).

Line 271 “for quality” instead of “fur quality”.

Response: Corrections have been made as recommended (Line 292).

Line 310 “which can effectively reduce the pH value in the intestine” have you measured the intestinal pH or that of the faeces ? again it would have been interesting to measure the quantity of lactic acid in the product.

Response: The pH values in the intestinal tract or in faeces were not measured. The measurement of lactic acid and short-chain fatty acids in the product were added in lines 84-86.

I hope our revision has addressed all the comments from you.  Please let me know if you have any questions or further comments.  Again, thank you for your value time!

Regards,

Yalin Li

College of Animal Science and Technology

Qingdao Agricultural University

Shandong, China

Round 2

Reviewer 1 Report

After making corrections and additions, I recommend your article for qualification for further editorial work.